# A Randomized Controlled Trial Evaluating Outcome Impact of Cilostazol in Patients with Coronary Artery Disease or at a High Risk of Cardiovascular Disease

**DOI:** 10.3390/jpm12060938

**Published:** 2022-06-06

**Authors:** Jia-Ling Lin, Wei-Kung Tseng, Po-Tseng Lee, Cheng-Han Lee, Shih-Ya Tseng, Po-Wei Chen, Hsien-Yuan Chang, Ting-Hsing Chao

**Affiliations:** 1Department of Internal Medicine, National Cheng Kung University Hospital, College of Medicine, National Cheng Kung University, Tainan 704302, Taiwan; jialingwing@hotmail.com (J.-L.L.); ptlee31@gmail.com (P.-T.L.); appollolee@hotmail.com (C.-H.L.); hero4811@gmail.com (S.-Y.T.); huntershobow@gmail.com (P.-W.C.); doyeric0926@yahoo.com (H.-Y.C.); 2Department of Internal Medicine, National Cheng Kung University Hospital, Dou-Liou Branch, College of Medicine, National Cheng Kung University, Yunlin 640003, Taiwan; 3Department of Internal Medicine, E-Da Hospital, Kaohsiung 824005, Taiwan; arthurtseng@me.com; 4Health Management Center, National Cheng Kung University Hospital, College of Medicine, National Cheng Kung University, Tainan 704302, Taiwan

**Keywords:** cilostazol, coronary artery disease, cardiovascular disease

## Abstract

Previous studies found that cilostazol has a favorable effect on glucose and lipid homeostasis, endothelial function, atherosclerosis, and vasculo-angiogenesis. However, it is poorly understood whether these effects can translate into better clinical outcomes. This study investigated the outcome effect of cilostazol in patients with coronary artery disease (CAD) or at a high risk of cardiovascular (CV) disease. We conducted a randomized, double-blind, placebo-controlled trial involving 266 patients who received cilostazol, 200 mg/day (*n* = 134) or placebo (*n* = 132). Pre-specified clinical endpoints including composite major adverse cardiovascular events (MACE) (CV death, non-fatal myocardial infarct, non-fatal stroke, hospitalization for heart failure, or unplanned coronary revascularization), the composite major coronary event (MCE) and major adverse CV and cerebrovascular event (MACCE), were prospectively assessed. The mean duration of follow-up was 2.9 years. Relative to placebo, cilostazol treatment had a borderline effect on risk reduction of MACE (hazard ratio [HR], 0.67; 95% confidence interval (CI), 0.34–1.33), whereas the beneficial effect in favor of cilostazol was significant in patients with diabetes mellitus or a history of percutaneous coronary intervention (*p* for interaction, 0.02 and 0.06, respectively). Use of cilostazol, significantly reduced the risk of MCE (HR, 0.38; 95% CI, 0.17–0.86) and MACCE (HR, 0.47; 95% CI, 0.23–0.96). A significantly lower risk of angina pectoris (HR, 0.38; 95% CI, 0.17–0.86) was also observed in the cilostazol group. After multi-variable adjustment, cilostazol treatment independently predicted a lower risk of MCE. In conclusion, these results suggest cilostazol may have beneficial effects in patients with CAD or at a high risk of CV disease.

## 1. Introduction

Cilostazol, a phosphodiesterase III inhibitor, provides antiplatelet and vasodilating effects by increasing the level of intracellular cyclic adenosine monophosphate [1]. It is involved in many cellular effects. Previously, we had performed a series of studies demonstrating the effects of cilostazol on angiogenesis via multiple pathways including the stromal cell-derived factor system, the adenosine monophosphate-activated protein kinase pathway, and extracellular-signal-regulated kinase/p38 mitogen-activated protein kinase pathways; thus, enhancing mobilization and proliferation of circulating endothelial progenitor cells [2,3,4,5,6]. Theoretically, the favorable effects of cilostazol on metabolic profile, endothelium-dependent functions, anti-atherosclerosis, and vasculo-angiogenesis imply that this compound might have prognostic benefits in patients with atherosclerotic cardiovascular (CV) disease [7,8,9]. However, it remains poorly understood whether these theoretical benefits can transform into better clinical outcomes.

Cilostazol is currently indicated for peripheral artery disease-related intermittent claudication [10]. Clinical use of cilostazol for other indications is being investigated. A recent meta-analysis of patients with peripheral artery disease showed that cilostazol treatment was associated with a lower risk of amputation and restenosis after either open or endovascular revascularization [11]; however, another meta-analysis showed no significant benefit of cilostazol regarding prevention of amputation [12]. In patients with cerebrovascular disease, some studies revealed that cilostazol could induce regression of carotid atherosclerosis, might prevent progression of symptomatic intracranial arterial stenosis, and even prevent stroke in patients with non-cardioembolic stroke [13,14,15]. For its use in coronary artery disease (CAD), clinical studies mainly focused on restenosis after percutaneous coronary intervention (PCI). Cilostazol might reduce the risk of restenosis of coronary stents [16], whereas other studies showed inconsistent effects of cilostazol on restenosis while comparing add-on cilostazol in patients already treated with aspirin and clopidogrel versus placebo [17,18,19,20]. There were also studies supporting this compound having outcome benefits in CAD patients by measuring surrogate endpoints [21,22,23]. When it comes to atherosclerotic CV disease, we should pay more attention to diabetic mellitus (DM), since DM is an important risk factor of atherosclerotic CV diseases, including CAD and other non-cardiac atherosclerotic diseases. Pharmacological treatment for this group of patients is of great interest, and tailored therapies are needed [24]. A recent review demonstrated that cilostazol might help to reduce diabetes-associated microvascular complications [25], but studies using cilostazol in patients with DM for macrovascular events were lacking.

Among the above potential benefits of cilostazol, to the best our knowledge, there has been no prospective, double-blind, randomized, controlled trial with a longer-term follow-up to evaluate the effects of cilostazol treatment on hard CV outcomes in patients with CAD or at a high risk of CV disease. Therefore, we conducted a prospective, double-blind, randomized, controlled trial to evaluate the effects of cilostazol on hard CV outcomes in patients with stable CAD or at a high risk of CV disease.

## 2. Materials and Methods

### 2.1. Study Design

This was a single-center, prospective, randomized, double-blind, placebo-controlled trial, done between 2014 to 2019, at National Cheng Kung University Hospital in Taiwan. Study protocol and amendments were approved by the Institutional Review Board of the National Cheng Kung University Hospital (IRB No. A-BR-102-076) and registered in www.ClinicalTrials.gov (https://clinicaltrials.gov/ct2/show/NCT02174939, accessed on 4 June 2022).

In a one-week run-in period, all eligible subjects were screened. Baseline blood samples were obtained. Then, we randomly assigned eligible participants to cilostazol 200 mg or placebo daily, using unrestricted randomization and sealed envelopes for allocation concealment. Participants, care providers, and outcome assessors were all blind to the assignment. Follow-up blood samples were obtained by the same procedure after 12-weeks of treatment.

### 2.2. Patient Population

We enrolled patients ≥20 years of age with stable CAD, including old myocardial infarction (MI) (>6 months), or at a high risk of CV disease. CAD was defined as patients who had both a positive exercise test and stress image test for ischemia in the last 12 months; or had significant luminal narrowing (≥50% diameter stenosis as compared to size of the adjacent reference vessel) in at least one of the coronary arteries (left main trunk, left anterior descending artery, left circumflex artery, and right coronary artery) or their major branches (diagonal branches, obtuse marginal branches, acute marginal branches, posterior descending artery, and posterior lateral branches) documented by dual-energy 128-row computed tomographic angiography or coronary angiography in the last 12 months. The diagnosis of MI was based on the 3rd version of the universal definition of MI [26]. High risk of CV disease was defined as patients who had pre-existing atherosclerotic CV disease other than CAD (including clinical or image evidence of peripheral artery disease or cerebrovascular disease), or at least one of the following situations: type 2 DM, metabolic syndrome, ≥stage 3 chronic kidney disease (CKD), and two or more coronary risk factors (male ≥ 45 years or female ≥ 55 years of age, tobacco smoking, hypertension, hyperlipidemia, family history of CV disease). Asian-modified metabolic syndrome was defined as individuals with three or more of the followings: waist size ≥ 90 cm in male or ≥80 cm in female; triglyceride ≥ 1.695 mmol/L (150 mg/dL); high density lipoprotein cholesterol < 1.036 mmol/L (40 mg/dL) in male or <1.295 mmol/L (50 mg/dL) in female; systolic blood pressure ≥130 mmHg or diastolic blood pressure ≥ 85 mmHg; fasting plasma glucose ≥ 5.5 mmol/L (100 mg/dL). Subjects were excluded if they had unstable CAD such as new onset angina (the last onset within eight weeks), crescendo angina, resting angina, and post-MI angina (the indexed infarction < 44 days), had plan to do percutaneous intervention or bypass surgery for CAD or peripheral artery disease within the last 3 months, severe liver dysfunction (transaminases ≥ ten times of upper normal limit, history of liver cirrhosis, or hepatoma), left ventricular dysfunction (ejection fraction ≤ 50% by echocardiography), documented active malignancy, chronic inflammatory disease, or current use of cilostazol or any other cyclic adenosine monophosphate elevator. Patients who had a known drug allergy history of cilostazol and premenopausal women were also excluded. Signed informed consent was obtained from all subjects involved in the study. The study was conducted according to the guidelines of the Declaration of Helsinki.

### 2.3. Outcomes and Follow-Up

Pre-specified clinical endpoints including the composite major adverse cardiovascular events (MACE) (CV death, nonfatal MI, nonfatal stroke, hospitalization for heart failure [HHF], and unplanned coronary revascularization), major coronary events (MCE) and major adverse cardiovascular and cerebrovascular events (MACCE) on long-term follow-up were prospectively assessed. The composite MCE included MI, angina pectoris, or unplanned coronary revascularization, while MACCE included CV death, nonfatal MI, nonfatal stroke, HHF, major amputation, minor amputation, or unplanned revascularization. Clinical outcomes were obtained by chart review and followed up by clinic visit, telephone call, or direct contact with participants or subjects’ family at 3 months after starting treatment and every 6 months thereafter.

### 2.4. Statistical Analysis

Distributions of continuous variables in both groups were expressed as mean ± standard error and skewed data were reported as median (interquartile range). Categorical variables were expressed as numbers and percentages. Mann–Whitney U test or unpaired Student’s t test were used for continuous variables while Chi-square or Fisher’s exact test were used for comparison of categorical variables between groups as appropriate.

Power estimation and sample size calculation were performed as follows. As previously reported, the TRANSCEND study (Telmisartan Randomised AssessmeNt Study in ACE iNtolerant subjects with cardiovascular Disease) showed a MACE (CV death, nonfatal MI, nonfatal stroke, and hospitalization for congestive heart failure, but no unplanned coronary revascularization) of 17% in the placebo group, at a median of 4.7 years after randomization [27]. The estimated MACE rate with placebo in the current study was 32% (unplanned coronary revascularization included). Accordingly, the sample size was calculated to observe a 14% absolute risk reduction in long-term MACE rate with a statistical power of 80% (1-beta) to detect a probability of 0.05 (alpha error level) estimated by using G*Power 3 software. Each analysis was performed by “intention-to-treat” and “per-protocol”. Pre-specified subgroup analysis was performed in subgroups such as old age (age ≥ 65 years), gender, CAD, DM, metabolic syndrome, hyperlipidemia, CKD (≥stage 3), and history of PCI. If more than one end point occurred within the follow-up period, only the first event was considered. Kaplan–Meier analysis was used to study patient survival and event-free status, using the log-rank test (Cox–Mantel) to ascertain differences between groups. A *p*-value < 0.05 was regarded as statistically significant. All statistical analyses were performed using SPSS 10.0 for Windows (SPSS Inc., Chicago, IL, USA).

## 3. Results

Initially, we enrolled 279 patients. A total of five were excluded due to screening failure and consent issues. In total, 274 patients were randomized into a 1:1 ratio to cilostazol and placebo. Later, eight patients withdrew consent. Finally, 134 patients assigned to cilostazol treatment, and 132 patients assigned to placebo were included in the intention-to-treat analysis despite 30 patients in the cilostazol group and 15 patients in the placebo group having discontinued the study drug. Figure 1 shows the CONSORT flow diagram of the study progress through all phases.

The average age was 64.5 years in the cilostazol group and 67.2 years in the placebo group. Males dominated in both groups (77.6% in cilostazol group and 68.9% in placebo group). Most of the baseline characteristics were similar between cilostazol and placebo groups, except participants in the cilostazol group were younger and more of them had peripheral artery disease, coronary bypass surgery, anemia, and use of aspirin. Most subjects had stable CAD (76.9% in the cilostazol group and 73.5% in the placebo group) and pre-existing atherosclerotic CV disease other than CAD. Therefore, participants with risk factors only were few (4.5% in the cilostazol group and 9.8% in the placebo group). Details of the baseline characteristics are listed in Table 1.

The mean duration of follow-up was 2.9 years and outcome results are shown in Table 2 and Figure 2. By intention-to-treat analysis, the composite endpoints of MACE occurred in 14 patients in the cilostazol group (10.4%) and in 20 patients in the placebo group (15.2%), leading to a borderline significant risk reduction in the primary composite endpoints (hazard ratio [HR], 0.67; 95% confidence interval [CI], 0.34–1.33, *p* = 0.25) with cilostazol treatment. For secondary composite endpoints, MCE occurred in eight patients in the cilostazol group (6.0%) and in 20 patients in the placebo group (15.2%) and MACCE occurred in 11 patients in the cilostazol group (8.2%) and in 22 patients in the placebo group (16.7%). Compared with placebo, cilostazol treatment significantly reduced the risk of MCE (HR, 0.38; 95% CI, 0.17–0.86, *p* = 0.02) and MACCE (HR, 0.47; 95% CI, 0.23–0.96, *p* = 0.04). For individual component of MACE, MCE, and MACCE, we observed a significantly lower risk of angina pectoris (HR, 0.38; 95% CI, 0.17–0.86, *p* = 0.02) and a trend toward a lower risk of unplanned revascularization (HR, 0.13; 95% CI, 0.02–1.09, *p* = 0.06) in the cilostazol group. The rest of the individual components did not differ significantly.

The outcome benefits of cilostazol observed in the intention-to-treat analysis were also present in the per-protocol analysis, with a borderline risk reduction in MACE (HR, 0.50; 95% CI, 0.22–1.16, *p* = 0.11) and a statistically significant risk reduction in MCE (HR, 0.24; 95% CI, 0.08–0.71, *p* = 0.01) and MACCE (HR, 0.36; 95% CI, 0.15–0.84, *p* = 0.02) (Appendix A). There was a significantly lower risk of angina pectoris (HR, 0.24; 95% CI, 0.08–0.71, *p* = 0.01) and a borderline lower risk of unplanned revascularization (HR, 0.15; 95% CI, 0.02–1.25, *p* = 0.08) in the cilostazol group.

By multivariable analysis, our data showed that cilostazol treatment was independently associated with a lower risk of MCE while some uni-variables were adjusted (HR, 0.34; 95% CI, 0.15–0.78, *p* = 0.01; Table 3), whereas this compound was not independently associated with a lower risk of MACCE (HR, 0.48; 95% CI, 0.23–1.02, *p* = 0.055; Appendix A). A pre-specified subgroup analysis of the effect of cilostazol versus placebo showed that DM subgroup might benefit from cilostazol treatment with respect to the occurrence of MACE (P for interaction, 0.02; Figure 3) and especially PCI subgroup could benefit from cilostazol treatment with respect to the occurrence of MCE and MACCE (*p* for interaction, 0.001 and 0.03, respectively; data not shown).

## 4. Discussion

The current study demonstrated, for the first time, that, although use of cilostazol was not associated with a significantly lower risk of a composite of CV death, nonfatal MI, nonfatal stroke, HHF, and unplanned coronary revascularization in patients with stable CAD or at a high risk of CV disease, the risks of a composite of MCE (MI, angina pectoris, or unplanned coronary revascularization) and MACCE (CV death, nonfatal MI, nonfatal stroke, HHF, major amputation, minor amputation, or unplanned revascularization) were significantly lower in cilostazol group, mainly driven by less angina pectoris and less unplanned revascularization.

Unlike some established pharmacotherapy for CV disease, such as statins, with large-scale trials demonstrating benefits in clinical outcomes, few studies have investigated the role of cilostazol in preventing CV events. To the best of our knowledge, our current study is the first randomized trial aimed at investigating the outcome benefits of cilostazol in terms of hard clinical endpoints for a longer-term follow-up in patients with stable CAD or at a high risk of CV disease. Although the numerical reduction in primary endpoint did not meet statistical significance, the preliminary findings in favor of cilostazol treatment from the current trial warrant further investigation in larger scale randomized controlled studies.

For patients with stable CAD, the well-known antithrombotic agent of choice is aspirin. Clopidogrel serves as an alternative for those with aspirin intolerance. The same recommendation goes for patients at a risk of CV disease despite that grade of recommendation being lower for primary prevention [28,29,30,31]. There is no evidence to support the clinical role of cilostazol in patients with CAD or at a high risk of CV disease prior to our current study. Cilostazol is generally used in this patient population under two clinical situations: either as an alternative to aspirin and clopidogrel for patients with classical antiplatelet intolerance, or as an add-on medication for patients requiring a stronger antithrombotic therapy. With respect to the first situation, studies from China showed that cilostazol is an effective and safe substitute of aspirin for patients undergoing coronary stenting [32,33]. Regarding the second situation, results from two meta-analyses of randomized trials comparing cilostazol triple therapy with standard dual antiplatelet treatment after PCI were inconsistent in terms of a risk reduction in MACE [34,35]. The ESCAPE study compared cilostazol with aspirin directly for primary prevention in patients who had a history of DM and 10–75% coronary stenosis by coronary computed tomographic angiography. After 12 months of treatment, cilostazol resulted in a non-significant decrease in coronary stenosis, while aspirin led to a non-significant increase in coronary stenosis [23]. The study population in the ESCAPE study was somewhat similar to our study; however, it included only subjects with DM and did not look into clinical hard outcomes.

Our sub-group analysis showed that cilostazol treatment was favored in patients with DM or those with a history of PCI. Clinical effects of cilostazol with respect to surrogate or soft clinical endpoints in both groups had been evaluated previously. Studies dedicated for patients with DM showed that cilostazol was beneficial in terms of favorable impact on platelet function, coronary stenosis or in stent restenosis rather than MACE [23,36,37,38]. Nevertheless, a meta-analysis comparing triple antiplatelet therapy with dual therapy in patients with DM after PCI reported that triple therapy was associated with a lower risk of MACE despite MACE not being the primary endpoint in these trials [39]. Recently, we and other researchers have found that this compound has beneficial effects on metabolic parameters and vasculo-angiogenesis functions in diabetes, either bench or bedside [1,2,40]. Taken together, the DM subgroup analysis from our study, along with the results of previous studies in diabetes, implied that cilostazol might be a good choice of antiplatelet agent for diabetic patients with CAD or at a high risk of CV disease. Since our study was not designed exclusively for patients with DM, this study was underpowered to conclude that cilostazol leads to better clinical outcomes in patients with DM. Further studies are needed to provide more robust evidence for use of cilostazol in these patients.

Furthermore, the PCI subgroup analysis from our current study was consistent with the findings from previous trials specifically focusing on patients undergoing PCI [18,21,41]. The outcome benefits were mainly driven by a risk reduction in revascularization [18,21]. Cilostazol has been reported to have a favorable effect on modification of coronary [23] or carotid plaque [42], and inhibition of neointimal hyperplasia after percutaneous intervention, either on coronary or peripheral arteries [16,43]. Besides, another individual outcome benefit of cilostazol in our study was less angina pectoris, probably owing to its vasodilatory effect. The STELLA trial showed that cilostazol significantly reduced chest pain frequency in patients with established diagnosis of vasospastic angina under amlodipine treatment [22]. Our study implied that, in addition to patients with vasospastic angina, those with stable CAD or at high risk of CV disease may also have better anginal control with cilostazol treatment.

There were some limitations in the current study. First, this was a single-center study in Asia. Our results cannot be generalized to other ethnicities. Secondly, despite the heterogeneity of the study population, the majority of the participants in the current study had CAD and the subgroup analysis showed consistent effects across patients with or without CAD. Thirdly, use of aspirin was not balanced between both groups. It is uncertain if this imbalance will interfere with the cilostazol effect on clinical outcome, although multi-variable analysis was performed. Fourthly, the discontinuation rate of cilostazol was higher than previous studies [18,21], probably leading to a neutral effect on MACE. Nevertheless, the per-protocol analysis also showed similar results. Finally, the observed event rate of MACE in the placebo arm was lower than expected and the calculated sample size was therefore relatively small, thereby leading to a neutral effect on MACE and reduced statistical power. Nevertheless, the beneficial effects of cilostazol treatment on secondary composite endpoints and other endpoints observed in the current trial warrant further investigation in larger scale randomized controlled studies.

## 5. Conclusions

In conclusion, cilostazol treatment might reduce CV risks in patients with stable CAD or at a high risk of CV disease; the beneficial effect in favor of cilostazol treatment appeared to be a hypothesis-generating signal in some subgroups, namely patients with DM or a history of PCI. Accordingly, larger scale CV outcome trials may provide further insight into the potential role of cilostazol in reducing CV events. Furthermore, studies focusing on patients with DM or a history of PCI may help prove the hypothesis that these groups are the most likely to benefit from cilostazol therapy.

## Figures and Tables

**Figure 1 jpm-12-00938-f001:**
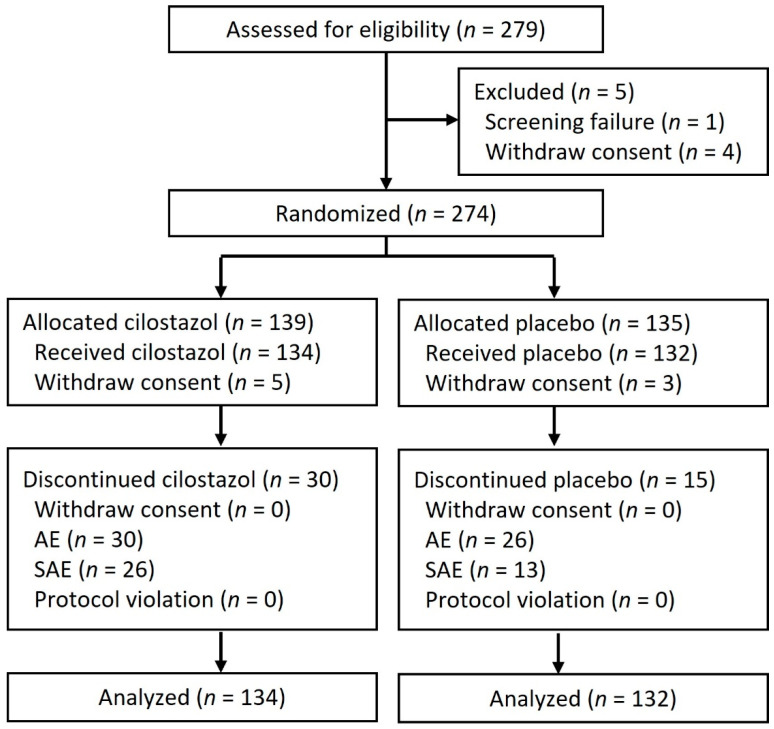
Participant flow diagram. AE, adverse event; SAE, severe adverse event.

**Figure 2 jpm-12-00938-f002:**
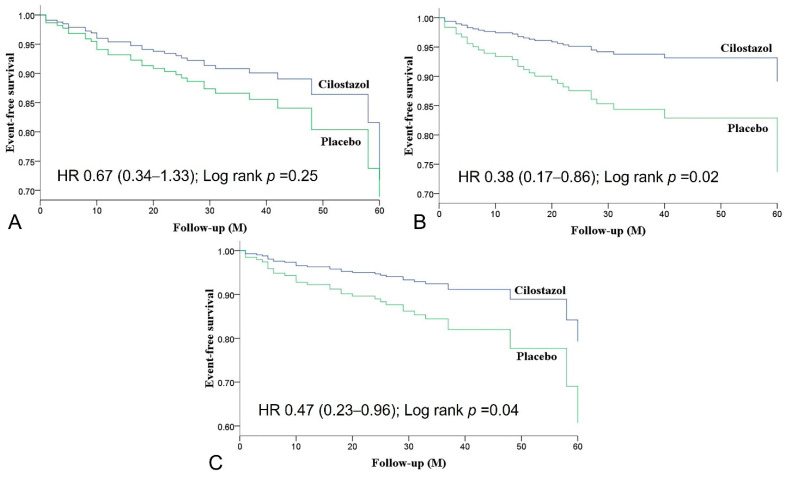
Event-free survival of the composite endpoints stratified by cilostazol or placebo use by Kaplan–Meier analysis using the log-rank test. (**A**) Major adverse cardiovascular events; (**B**) Major coronary events; (**C**) Major adverse cardiovascular and cerebrovascular events. HR, hazard ratio; M, months.

**Figure 3 jpm-12-00938-f003:**
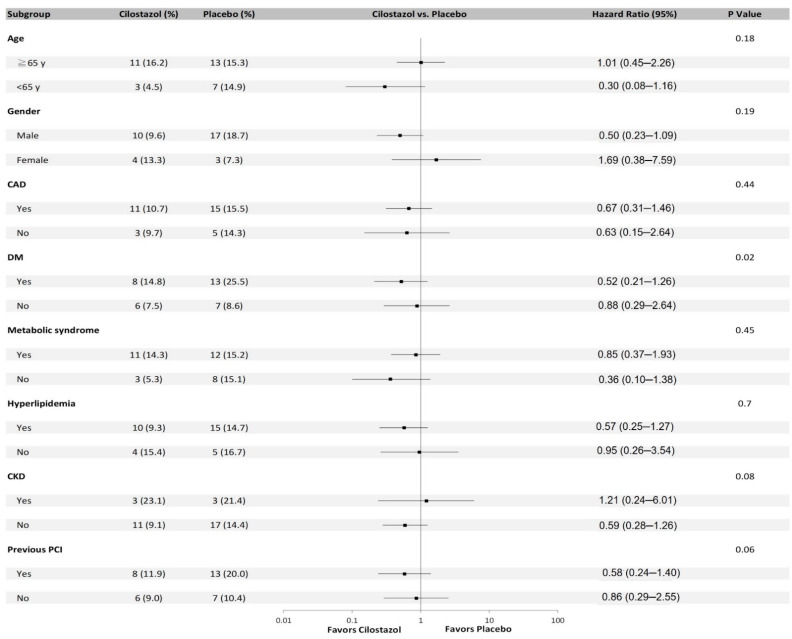
Subgroup analysis of the effect of cilostazol versus placebo on the occurrence of the composite major adverse cardiovascular events.

**Table 1 jpm-12-00938-t001:** Comparisons of the baseline characteristics between participants assigned to cilostazol and placebo.

	Cilostazol (*n* = 134)	Placebo (*n* = 132)	*p*-Value
Age, year	64.5 (9.6)	67.2 (9.9)	0.03
Male gender	104 (77.6)	91 (68.9)	0.11
Indications			
Stable CAD	103 (76.9)	97 (73.5)	0.52
Old MI	38 (28.4)	29 (22.0)	0.23
High risk of CV disease	31 (23.1)	35 (26.5)	0.52
Risk factors only	6 (4.5)	13 (9.8)	0.09
Underlying disease			
Diabetes mellitus	54 (40.3)	51 (38.6)	0.78
Hypertension	101 (75.4)	106 (80.3)	0.33
Hyperlipidemia	108 (80.6)	102 (77.3)	0.51
Metabolic syndrome	77 (57.5)	79 (59.8)	0.69
Tobacco smoking	30 (22.4)	27 (20.5)	0.70
Chronic kidney disease	13 (9.7)	14 (10.6)	0.81
Anemia	24 (18.2)	41 (31.1)	0.02
CAD	103 (76.9)	97 (73.5)	0.52
MI	38 (28.4)	29 (22.0)	0.23
Peripheral artery disease	27 (20.1)	14 (10.6)	0.03
Cerebrovascular disease	8 (6.0)	5 (3.8)	0.40
PCI	67 (50.0)	65 (49.2)	0.90
CABG	6 (4.5)	0	0.03
PTA	0	0	
Medication			
Aspirin	94 (70.1)	77 (58.3)	0.04
Clopidogrel	12 (9.0)	19 (14.4)	0.17
Ticagrelor	1 (0.7)	2 (1.5)	0.62
ACEi	37 (27.6)	31 (23.5)	0.44
ARB	47 (35.1)	51 (38.6)	0.55
Statin	76 (56.7)	66 (50.0)	0.27
Objective data			
Blood pressure, mmHg			
Systolic	132.1 (13.0)	132.2 (14.2)	0.95
Diastolic	76 (70–84)	74 (70–83.75)	0.24
Creatinine, mg/dL	0.87 (0.75–1.11)	0.87 (0.71–1.07)	0.30
eGFR, ml/min/1.73 m^2^	84 (64–90)	82.69 (66.25–90)	0.86
Hemoglobin A1C, %	6.15 (5.7–7.2)	6.1 (5.8–7.0)	0.77
LDL cholesterol, mg/dL	108.0 (30.2)	109.7 (30.3)	0.63

Data are presented as number (percentages), median (interquartile range), or mean (standard deviation). ACEi, angiotensin converting enzyme inhibitor; ARB, angiotensin receptor blocker; ASCVD, atherosclerotic cardiovascular disease; CABG, coronary artery bypass graft surgery; CAD, coronary artery disease; CV, cardiovascular; eGFR, estimated glomerular filtration rate; LDL, low-density lipoprotein; MI, myocardial infarction; PCI, percutaneous coronary intervention; PTA, percutaneous transluminal angioplasty.

**Table 2 jpm-12-00938-t002:** Comparisons of the composite endpoints and individual endpoints between participants assigned to cilostazol and placebo (intention-to-treat analysis).

	Cilostazol (*n* = 134)	Placebo (*n* = 132)	HR (95% CI)	*p*-Value
Composite endpoints				
MACE	14 (10.4)	20 (15.2)	0.67 (0.34–1.33)	0.25
MCE	8 (6.0)	20 (15.2)	0.38 (0.17–0.86)	0.02
MACCE	11 (8.2)	22 (16.7)	0.47 (0.23–0.96)	0.04
Individual endpoints				
CV death	2 (1.5)	2 (1.5)	0.99 (0.14–7.00)	0.99
Nonfatal MI	1 (0.7)	4 (3.0)	0.23 (0.03–2.09)	0.19
Nonfatal stroke	5 (3.7)	5 (3.8)	0.93 (0.27–3.23)	0.91
HHF	3 (2.2)	7 (5.3)	0.42 (0.11–1.62)	0.21
Angina pectoris	8 (6.0)	20 (15.2)	0.38 (0.17–0.86)	0.02
Unplanned coronary revascularization	5 (3.7)	8 (6.1)	0.60 (0.20–1.83)	0.37
Unplanned revascularization	1 (0.7)	7 (5.3)	0.13 (0.02–1.09)	0.06
Major amputation	0	0		
Minor amputation	0	0		

CI, confidence interval; CV, cardiovascular; HHF, hospitalization for heart failure; HR, hazard ratio; MACCE, major adverse cardiovascular and cerebrovascular event; MACE, major adverse cardiovascular event; MCE, major coronary event; MI, myocardial infarction.

**Table 3 jpm-12-00938-t003:** Uni- and multi-variables independently predicting major coronary events.

Variables	Uni-Variables HR (95% CI)	*p*-Value	Multi-Variables HR (95% CI)	*p*-Value
Cilostazol	0.38 (0.17–0.86)	0.02	0.34 (0.15–0.78)	0.01
Age	1.01 (0.97–1.05)	0.56		
Peripheral artery disease	0.89 (0.32–2.48)	0.82		
Coronary bypass surgery	0.05 (0–10296.19)	0.63		
Anemia	0.93 (0.37–2.31)	0.88		
Aspirin	2.52 (0.96–6.63)	0.06	2.88 (1.09–7.61)	0.03

CI, confidence interval; HR, hazard ratio.

## Data Availability

The data presented in this study are available on request from the corresponding author. The data are not publicly available due to ethic and privacy regulation.

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
