# Peer review of "A Randomized Controlled Trial Evaluating Outcome Impact of Cilostazol in Patients with Coronary Artery Disease or at a High Risk of Cardiovascular Disease"

_jpm, 2022, doi:10.3390/jpm12060938_

Round 1
Reviewer 1 Report
This paper is very interesting. Introduction could be improved a little. Please add in the introduction also place i DM patient, please see doi: 10.1177/1074248420941675
Study is very well designed. Results and discussion are well presented.
Tables /Figures - concise and adequately present results.
Conclusion need to be a little expanded, e.g. add more specifically future ways ii this field.
Reviewer 2 Report
Lin et al have assessed the use of cilostazol treatment in patients with CAD. This is a very clear and carefully designed study with promising outcome.
Incidence of CAD/CVD is many fold higher in patients with metabolic disease, especially diabetes. It is unclear in this study if there were any direct correlations were made between diabetic and non-diabetic patients with CVD, for instance, did cilostazol treatment showed better clinical outcome in diabetics ?
On another note, does cilostazol treatment have more or similar treatment benefits like statins ? Discussing this will help highlight the potential of this drug.
